# Oil Deposition on Polymer Brush-Coated NF Membranes

**DOI:** 10.3390/membranes9120168

**Published:** 2019-12-06

**Authors:** Anh Vu, Naama Segev Mark, Guy Z. Ramon, Xianghong Qian, Arijit Sengupta, S. Ranil Wickramasinghe

**Affiliations:** 1Ralph E Martin Department of Chemical Engineering, University of Arkansas, Fayetteville, AR 72701, USA; anhtvu1985@gmail.com (A.V.); arijitbarc@gmail.com (A.S.); 2Department of Civil & Environmental Engineering, Technion—Israel Institute of Technology, Haifa 32000, Israel; naamasegev88@gmail.com (N.S.M.); ramong@technion.ac.il (G.Z.R.); 3Department of Biomedical Engineering, University of Arkansas, Fayetteville, AR 72701, USA; xqian@uark.edu

**Keywords:** direct observation, emulsion, responsive membrane, thermo-responsive polymer, surface modification, wastewater

## Abstract

Membrane-based processes are attractive for treating oily wastewaters. However, membrane fouling due to the deposition of oil droplets on the membrane surface compromises performance. Here, real-time observation of the deposition of oil droplets by direct confocal microscopy was conducted. Experiments were conducted in dead-end and crossflow modes. Base NF 270 nanofiltration membranes as well as membranes modified by grafting poly(N-isopropylacrylamide) chains from the membrane surface using atom transfer radical polymerization were investigated. By using feed streams containing low and high NaCl concentrations, the grafted polymer chains could be induced to switch conformation from a hydrated to a dehydrated state, as the lower critical solution temperature for the grafted polymer chains moved above and below the room temperature, respectively. For the modified membrane, it was shown that switching conformation of the grafted polymer chains led to the partial release of adsorbed oil. The results also indicate that, unlike particles such as polystyrene beads, adsorption of oil droplets can lead to coalescence of the adsorbed oil droplets on the membrane surface. The results provide further evidence of the importance of membrane properties, feed solution characteristics, and operating mode and conditions on membrane fouling.

## 1. Introduction

Membrane fouling compromises the performance of all membrane-based separation processes. Rejected and unwanted species can adsorb on the membrane surface. One strategy to minimize the effect of fouling is to modify the surface of the membrane in order to reduce adsorption of unwanted species on the membrane surface [1,2]. In this way, desirable bulk membrane properties, such as pore size, pore size distribution, and morphology, are preserved while tuning membrane surface properties. The present work focused on the fouling of nanofiltration (NF) membranes by oil–water emulsions. Developed initially for softening of surface and ground water [3], today, these membranes are used for many other applications, one of which is treating oily wastewaters. There are many sources of oily wastewater, the largest being from oil and gas production operations [4]. Depending on the source of the oily wastewater, many other contaminants can be present such as dissolved salts, surfactants, polar organic compounds, etc. [5]. Designing membrane surfaces that suppress fouling is challenging.

Here, we focused on grafting temperature responsive polymer brushes from the membrane surface. Previous studies have suggested that grafting stimuli-responsive polymer chains to the surface of the membrane can not only suppress fouling but also help promote the release of adsorbed foulants during cleaning cycles [6]. These stimuli-responsive polymers change their conformation in response to changed external conditions [7]. Here, we focused on polymer chains that exhibit a lower critical solution temperature (LCST). Specifically, we grafted poly(*N*-isopropylacrylamide) (PNIPAM) from the membrane surface.

Prior work indicated that grafting thermo-responsive polymers to the membrane surface can lead to “self-cleaning” membranes. For example, grafting PNIPAM chains from the surface of the membrane can promote the release of adsorbed species by cycling the temperature above and below its LCST. This leads to a reversible change in the conformation of the grafted nanostructure which could lead to the release of adsorbed species [8,9,10]. In recent studies, Zhang et al. [11], Wu et al. [12], and Wang et al. [13] exploited the changes in conformation of the grafted nanostructure above and below the LCST in order to develop membranes suitable for oil–water separations. Zhu and Guo [14] reviewed the development of membranes with tunable wettability, achieved via changes in the conformation of the membrane surface nanostructure above and below the LCST.

Predicting fouling for a given feed stream, membrane, and operating conditions is complex. The observed fouling depends on the interplay between feed solution and membrane properties as well as operating conditions [15]. Frequently, autopsies are conducted in order to analyze the foulants on the membrane surface. However, these studies provide little information on adsorption and release during filtration.

Xu et al. [16] used ultrasonic reflectometry and wavelet analysis to visualize fouling and diffusion during microfiltration of oily wastewater. Their results provide insights into the onset and location of adsorption onto the membrane surface. Tummons et al. [17] used direct observation through the membrane for real-time analysis of membrane fouling. They identified three stages of membrane fouling: (1) droplet attachment and clustering; (2) droplet deformation; (3) droplet coalescence. Tanudjaja et al. [18] extended this work by using direct observation through the membrane to investigate the differences in fouling by colloidal particles and oil–water emulsions. They noted that the critical flux for oil droplets and latex particles of the same diameter was very different. Fux and Ramon [19] used confocal microscopy for direct observation of oil droplet deposition, deformation, and detachment during separation and cleaning, respectively. They observed that operation at a low flux yields spherical droplets that are easily removed by crossflow cleaning, whereas a high flux leads to significant deformation and mostly irreversible deposition [19].

Lin and Rutledge [20] used direct observation through the membrane to observe fouling of electrospun membranes when challenged with oil-in-water emulsions. Tanudjaja and Chiew [21] used direct observation through the membrane to quantify the critical flux for a number of different oils in water solutions. They compared fouling predictions by the DLVO and XDLVO models. Tanis-Kanbur et al. [22] also used direct observation through the membrane to observe the onset of fouling and determine critical flux. They combined these experimental results with molecular dynamics simulations to understand interfacial interactions and the mechanism of fouling. Both of these studies indicate the importance of interactions among oil droplets.

In our earlier work, we investigated colloidal deposition on PNIPAM-coated NF membranes [23]. We used direct confocal microscopy imaging to investigate, in real time, deposition of colloidal particles as well as the effect of changing the conformation of the grafted PNIPAM. The PNIPAM had an LCST at 32 °C in DI water. However, at high ionic strength, the LCST dropped to ~20 °C. Consequently, we were able to use a high concentration NaCl followed by DI water solution at room temperature (~25 °C) to switch above and below the LCST of PNIPAM. In the case of colloidal fouling, we noted that grafting PNIPAM chains from the surface of the base membrane suppressed adsorption of colloidal particles relative to the unmodified base membrane. The degree of adsorption was affected by factors such as surface charge—if the particles and membrane surface were oppositely charged, adsorption was greater. Interestingly, our results indicated that, once adsorbed, colloidal particles could not be released by switching the polymer conformation.

The present contribution extends our previous studies. Our aim was to observe deposition and release of oil droplets from the membrane surface which was modified by grafting thermo-responsive polymer brushes. Based on the substantial differences observed in earlier studies between colloidal fouling and fouling by oil–water emulsions, we investigated the effect of PNIPAM chains grafted from the surface of an NF membrane on fouling by oil droplets in water. Using direct microscopic visualization, we studied both adsorption of the oil droplets onto the membrane surface as well as the effect of switching the PNIPAM conformation by adding high concentration NaCl to the feed stream.

## 2. Materials and Methods

### 2.1. Materials

All reagents were ACS grade unless otherwise specified. Acetonitrile and methanol were purchased from EMD Millipore, Billerica, MA, USA, while ethanol was purchased from VWR International, Radnor, PA, USA. N-Isopropylacrylamide (>98%) was purchased from TCI America, Boston, MA, USA. α-Bromoisobutyryl bromide (98%) was purchased from Alfa–Aesar, Ward Hill, MA, USA. Triethylamine, 4-dimethylaminopyridine, *N*,*N*,*N*’,*N*”,*N*”-pentamethyl diethylenetriamine (99% (PMDETA)), sodium chloride, CuBr, and CuBr_2_ were purchased from Sigma–Aldrich, Munich, Germany. The deionized water used throughout the investigation was obtained from a Thermo Fisher 18 MΩ Barnstead Smart2Pure system, Schwerte, Germany. The NF 270 membranes were provided by Dow Filmtec, Edina, MN, USA. The membranes were cut into 25 mm discs.

### 2.2. PNIPAM Grafting

A two-step surface modification method was used to graft the PNIPAM from the surface of the NF 270 membranes. The first step was initiator immobilization, while the second step was atom transfer radical polymerization (ATRP). All the membrane samples were washed thoroughly in a 1:1 (*v*/*v*) ethanol–water mixture for 2 h, followed by drying overnight in a vacuum oven at 40 °C. Initiator immobilization was carried out in acetonitrile. The reaction mixture was prepared by dissolving 100 mM triethylamine and 5 mM 4-dimethylaminopyridine in acetonitrile. The NF 270 membrane discs were added to the solution. Then 1% by volume of α-bromoisobutyryl bromide (ATRP initiator) was added to the reaction mixture.

Initiator immobilization was carried out for 15 min at ambient temperature followed by quenching with water. Next, the membranes were washed with a 1:1 (*v*/*v*) ethanol–water mixture to remove unreacted precursors and then were dried overnight [24,25].

The ATRP was carried out in a three-necked, round-bottom flask. A homogeneous reaction mixture containing NIPAM:CuBr:PMDETA in a molar ratio of 100:1:3 in a 9:1 (*v/v*) water/methanol solvent was used. Initially, a solution of NIPAM in water–methanol solvent was prepared by mixing the components for 30 min in an argon atmosphere. Next, PMDETA was added under continuous stirring for 15 min. The membrane was then added to the reaction mixture followed by addition of CuBr under an argon atmosphere. Polymerization times investigated here were 1–4 h. Finally, the membranes were washed thoroughly with a water/methanol 1:1 (*v*/*v*) mixture to remove unreacted species. The membranes were stored in a water/ethanol 1:1 (*v*/*v*) mixture. Figure 1 schematically presents the modification process.

### 2.3. Membrane Characterization

Attenuated total reflectance Fourier transform infrared spectroscopic (ATR-FTIR) analysis was carried out using a Shimadzu IRAffinity-1 spectrometer equipped with a PIKE single-reflection horizontal ATR accessory procured from Shimadzu, Columbia, MD, USA. A wide spectral range of 600–4000 cm^−1^ was investigated to obtain the optimum signal-to-noise ratio.

The water contact angle on the membrane surface was measured in static mode using the captive bubble method at room temperature and pressure (OCA 20, Future Digital Scientific Corp., Garden City, NY, USA). The membranes were dried overnight prior to analysis. Using the circle fitting method, the angle between the air bubble and the membrane surface was measured every 0.1 s at 5 different locations on the surface. The surface charge on the membrane was measured in terms of zeta potential using a Delsa Nano HC particle analyzer (Beckman Coulter, Brea, CA, USA).

Atomic force microscopic (AFM) analysis of the membrane surface was conducted using an atomic force microscope (AFM) equipped with a Dimension Icon AFM from Bruker, Santa Barbara, CA, USA. NanoScope V815R3sr1 software with the NanoScope Analysis program were used to analyze the collected data. The membrane samples were dried overnight prior to analysis.

The degree of grafting was determined for all modified membranes. The base membrane was rinsed and dried overnight in a vacuum oven at 40 °C. The dried weight of the unmodified membrane was recorded. After modification, the membrane was washed in DI water and then dried overnight in a vacuum oven at 40 °C. The membrane was then weighed again. The degree of grafting (DG, mg/cm^2^) was calculated using the following equation:(1)$$DG=\frac{\left(W_{1}\;-\;W_{0}\right)}{A}$$where *W*_0_ is the mass of the unmodified membrane and *W*_1_ is the mass of the membrane after modification and drying. *A* is the membrane area,13.4 cm^2^ in this study.

### 2.4. Membrane Performance

#### 2.4.1. Emulsion Preparation 

To achieve a 50 μL/L concentration, a volume of 25 μL of hexadecane oil was added to 50 mL of deionized water and stained with 1% Dye-Lite (Fluorescence Dye, 561 nm excitation). Then, the emulsion was sonicated for 3 min at room temperature (20 °C). After sonication, the emulsion was added to 450 mL of deionized water mixed with 2.5 μL Triton-X-100 surfactant (1:10 surfactant/oil (*v*/*v*) ratio) and stirred on a magnetic plate for 10–15 min. The oil droplet size was determined using a Mastersizer (Malvern Instruments 2000, Malvern, UK). Droplet size distribution ranged from 1 to 30 µm with most of the droplets (50%) with a mean of 4 µm. The oil droplets’ zeta potential in the presence of Triton-x-100 was measured using Zetasizer (Malvern Instruments, Malvern, UK) at three pH values: 4, 6, and 10, and two sodium chloride concentrations: 0 and 0.005 M. At pH 6, the zeta potential of the emulsion was −44.5 and −16.3 mV for 0 and 5 mM, respectively.

#### 2.4.2. Experimental System 

A custom-made crossflow filtration cell, with a sapphire glass viewing port, allowed in situ observation using a confocal laser scanning microscope (TCS SP8, Leica Microsystems, Wetzlar, Germany). An image of the system is shown in Figure 2. Channel dimensions within the cell were 0.6 mm (H), 6 mm (W), and 36 mm (L) with a total membrane filtration area of 216 mm^2^. The flow cell was mounted on the microscope stage equipped with a 25xX water immersion objective, corresponding to a field of view of 262,144 pixels (384,400 μm^2^). Experiments were carried out at room temperature (22 °C) and pH 6 (unadjusted). The feed vessel was placed on top of a magnetic plate and the solution was stirred at 400 rpm. The applied feed pressure was set to 3 bar, using compressed nitrogen connected to the feed vessel. A programmable gear pump (Micropump, Cole Parmer, Vernon Hills, IL, USA) was used to maintain a constant crossflow velocity of 0.07 m/s and which recirculated the retentate back to the feed vessel.

Experiments were performed at two operational modes: crossflow and dead-end filtration. The permeate outlet was connected to a syringe pump which was set at a constant flow rate to ensure a constant flux through the membrane. Changes in the transmembrane pressure (TMP), occurring during the experiment due to the fouling, were monitored by a differential pressure transducer (PX409, Omega Engineering, Manchester, UK). Electrical conductivity and temperature were measured continuously with a probe (M200 easy line, Mettler Toledo, Lutz, FL, USA) dipped into the feed vessel. Electrical conductivity of the permeate was measured at the end of each experiment by a probe (CyberScan PC 300, Eutech Instruments, Singapore).

#### 2.4.3. Deposition and Cleaning Experiments 

Initially, air bubbles were removed from the system, and the membrane was stabilized before each experiment. Feed solution was pumped into the flow cell for 40 min for deposition, during which five random images of the membrane surface at different locations were acquired every 5 min. Deposition was followed by a cleaning protocol composed of 5 steps. First, the feed pressure was reduced to zero (designated as TMP = 0) to eliminate permeation (while the gear pump still operated). Then, the feed solution was replaced with a concentrated salt solution containing 1.5 M NaCl (salt), after which the feed was replaced by deionized water (water). The last two steps were repeated twice. Each of the cleaning steps lasted 20 min. After the “cleaning cycles” were completed, the flux recovery was determined with DI water. The average permeation rate at a pressure of 3 bar was 17.7 and 3.6 µm/s for NF 270 and PNIPAM, respectively. To overcome this difference in conditions, experiments were performed under a larger oil concentration for PNIPAM (so as to be conservative); the oil concentration in the PNIPAM experiments was almost 4 times higher than that for the base NF 270. The oil flux towards the base membrane was 8.8 ×·10^−13^ µl(oil)/s·µm^2^ and for PNIPAM modified membrane, 2.4 × 10^−13^ µl(oil)/s·µm^2^. The oil flux was calculated according to *J*_w_ = *J*_o_
*C*_o_, where *J*_o_ is the oil flux towards the membrane (µl(oil)/s·µm^2^), *C*_o_ is the oil concentration (µl(oil)/µm^3^), and *J*_w_ is the water flux through the membrane (μm/s).

#### 2.4.4. Image Acquisition and Analysis

The high-resolution images acquired through the microscope were exported to a PC and analyzed with MATLAB (version R2016a). Each image was represented in pixels by modifying it to 8 bit integers. In a raw gray-scale image, each pixel holds 256 steps of gray level values which describe the intensity (0 = black, 255 = white). Fluorescent oil droplets appeared white against the dark membrane background. Then, a threshold was chosen to eliminate background noise in the images, and the converted black and white images were used, via a pixel enumeration procedure, to determine the fractional coverage by oil at a given time.

## 3. Results and Discussions

### 3.1. Membrane Characterization

The FTIR spectra of the base and modified membrane are shown in Figure 3a. As can be seen after modification, a peak at 1635 cm^–1^ is observed which can be assigned to the amide functionality due to the PNIPAM grafting [26]. Figure 3b gives the water contact angle for the base and modified membrane. Grafting PNIPAM from the membrane surface increased the water contact angle, indicating an increase in the hydrophobicity of the membrane surface. Finally, Figure 3c gives the zeta potential for the base and modified membranes. At pH 7, the zeta potential of the unmodified NF 270 membrane was −15 mV which was reduced to −10 mV upon grafting of PNIPAM from the membrane surface. This result is consistent with the observed change in the contact angle after surface modification.

Figure 4 shows the results of 2D and 3D AFM images for the base and modified membrane which clearly indicate the change in surface morphology after grafting poly(NIPAM). A significant increase in surface roughness was observed. The surface roughness for the base membrane was 3.42 nm and increased to 6.20 nm for the modified membrane. This result agrees with earlier studies [27,28].

Figure 5 gives the variation in the degree of grafting with reaction time. A linear increase in grafting degree with time indicates a highly controlled polymerization reaction. The ATRP is a controlled grafting method, where there is a dynamic equilibrium between Cu(I) and Cu(II). 

### 3.2. Oil Deposition Studies

Our recent investigation revealed that, even though there was an enhancement in surface hydrophobicity due to the grafting of PNIPAM on NF 270, it hindered the deposition of foulant particles regardless of their charge, resulting in improved fouling resistance for colloids. Here, microscopic observation was used in order to understand the deposition and release patterns of oil droplets fouling on PNIPAM-grafted NF 270 membranes. 

Figure 6 shows the variation in water flux as a function of filtration time for the base and modified membrane. Though we investigated polymerization times varying for 1–4 h, the permeate flux at longer polymerization times was significantly reduced. In order to compare deposition results for base and modified membranes, it is important to ensure similar permeate fluxes. Consequently, deposition experiments were conducted for membranes modified using 1 h polymerization times only. Initially, the membranes were challenged with DI water. Next, feed streams consisting of increasing concentrations of NaCl were introduced. In the case of the base membrane, there was an associated step change in flux. The flux dropped as the NaCl concentration increased due to the increased osmotic back pressure. The response of the modified membrane was very different. As the salt concentration increased, the LCST decreased until it was below room temperature (operating temperature). Below the LCST, PNIPAM exists as a brush structure with extensive hydrophilic interaction with the water molecules. However, above the LCST, the polymers attain a hydrophobic coiled conformation, generating an additional hydrophobic barrier for passage of water molecules. The slow decrease in the flux of the modified membrane, compared to the more abrupt flux drop of the base membrane, indicates the conformational changes of PNIPAM as the concentration of NaCl increases. Finally, the membranes were challenged with DI water again. For both membranes, the flux returned close to the initial DI water flux. This indicates the reversibility of the change in PNIPAM conformation above and below its LCST. Full flux recovery was not achieved, possibly due to the compaction during filtration. In our previous work [27], we showed that is important not to dry the membranes during storage after testing. As long as the membranes are not completely dried and there is no irreversible fouling, switching between the hydrated and collapsed states is completely reversible.

The percentage surface coverage by oil droplets of base NF 270 and PNIPAM-modified NF 270 membranes is shown in Figure 7. The initial 40 min of operation was in dead-end mode and represents the deposition stage of the oil droplets. The degree of the surface coverage for base and modified membranes was the same. Next, the pressure was reduced to zero for a period of 40–60 min. As can be seen, the surface coverage for the unmodified membrane reduced to ~2%, while for the modified membrane, it decreased only ~13.5%, a minor reduction of 0.5%.

As shown in the membrane characterization results, the modification of PNIPAM slightly changed the base membrane surface properties, generating a rougher surface which was less hydrophilic and negatively charged. The significant release of droplets from the NF 270 surface implies that the droplets were held stationary mainly by permeation rather than adhesion to the surface. In terms of fouling removal, while there was no permeation, back migration of the oil droplets dominated. On the other hand, for the modified membrane, oil droplets were still covering the membrane surface even when there was no permeation. Since the membranes and the emulsion were both negatively charged (according to zeta potential measurements), this may be the outcome of hydrophobic interactions between oil droplets and PNIPAM chains.

Next, the membranes were flushed with two cycles of 1.5 M NaCl solution for 20 min, followed by DI water for 20 min. The pressure was kept at 0 during these cycles. These cycles of adjusting the LCST above and below the operating temperature of the system led to almost complete removal of oil from the base NF 270 membrane. For the modified membrane, the situation was very different. During the first NaCl exposure (65–80 min), the surface coverage was reduced by 3.5% (ANOVA *p*-value = 0.002) and by 2.5% (ANOVA *p*-value = 0.003) in the second exposure (105–120 min), i.e., when the LCST of PNIPAM was below the operating temperature. As the difference in the surface properties between the two membranes was modest, this result was the outcome of the conformational changes of the PNIPAM chains. Furthermore, it means that the deposited oil droplets were reversible. Later, when the PNIPAM chains were exposed to DI water, a slight increase in surface coverage was observed. This increase may be related to residual droplets that were left in bulk or to the large variation among the experiments (as indicated by the large error bars). This variation may also be the result of non-uniform droplet sizes. The overall tendency of this system was to release oil droplets from the surface throughout the cleaning process. 

The surface coverage (%) by oil droplets for the base NF 270 (red) and PNIPAM-modified (blue) membranes during crossflow filtration is given in Figure 8. The feed streams used were identical to the dead-end experiments (see Figure 7). As expected, during the first 40 min of operation the deposition was much greater for both membranes during dead-end operation. In the case of crossflow filtration, the surface coverage was reduced by approximately 50% when the transmembrane pressure was reduced to zero for the base and modified membranes. 

The first time the base membrane was exposed to a solution of 1.5 M NaCl, the surface coverage was reduced to 1.3%–1.4% and stayed almost constant to the end of the two high/low salt cycles. As for the PNIPAM-modified membrane, the surface coverage was reduced each time the membrane was exposed to 1.5 M NaCl, while when exposed to DI water, the surface coverage did not change. Eventually, the surface coverage reached ~1.3%. The significant statistical variance between the two membranes was confirmed by ANOVA test for all results. While the differences were not large, the trends observed for dead-end filtration were kept during crossflow filtration. Since the surface property differences between the membranes were humble (see Section 3.1.), once again, it can be stated that the effect of the conformational changes of the chains responsible for surface coverage decreased. The reduction in flux for the base membrane when challenged with NaCl solutions was due to the increase in osmotic back pressure. The changes in the flux for the PNIPAM-modified membrane were different than the base membrane. The results suggest that changes in the conformation of the PNIPAM chains were responsible for the observed changes in performance.

The three-dimensional images taken by the confocal microscope revealed some additional data about the size and shape of the deposited droplets (Figure 9). Comparing these results to our previous study on deposition of polystyrene beads suggests the differences between fouling by oil droplets and polystyrene beads. Polystyrene beads, once deposited, can generally not be removed by physical rinsing nor, in the case of the PNIPAM-coated membranes, by switching polymer conformation. Oil droplets, on the other hand, are flexible, deformable, and hydrophobic by nature; therefore, they behave differently. Besides the fact that the oil emulsion contained a range of droplet sizes (1–30 µm), the oil droplets tended to coalesce at the bulk or near the surface, contributing to additional changes in their shape and size. This process of coalescence may result in larger droplets or, even in extreme cases, the creation of an oil layer that may completely block the membrane pores. Furthermore, as the rejected oil droplets grow, as was shown by Fux and Ramon [19], they are more likely to deform and, thus, are less removable, at least within the size range examined (up to a 15 micron diameter). As seen in Figure 9, this phenomenon was still valid on different surface morphologies where large droplets “survive” the NaCl–DI water feed streams that change the conformation of the grafted PNIPAM chains.

The results obtained here provided real-time observation of the deposition of oil droplets on the membrane surface. Coalescence of the deformable oil droplets can lead to the formation of an oil layer on the membrane surface. Grafting PNIPAM chains from the membrane surface leads to different levels of surface coverage between the modified and base NF 270 membranes. These results provide further evidence of the importance of membrane properties, solution characteristics, and operating conditions on membrane performance.

## 4. Conclusions

Direct microscopic observation of oil droplet deposition on the PNIPAM-modified membranes offers new insights into oil–PNIPAM chain-coated membrane interactions. Hexadecane emulsion was used. Flux measurements of the PNIPAM-modified membranes demonstrated reversible conformational changes of the chains in the presence of a concentrated sodium chloride solution. Crossflow and dead-end filtration experiments, performed under confocal microscopy, indicated that the PNIPAM-modified membrane significantly decreases oil droplet deposition, apparently by transforming from a hydrophilic to hydrophobic state. 

Oil droplet coverage during deposition onto the base NF 270 membrane and PNIPAM-modified membrane was quite similar. Unlike polystyrene beads, reversibility of the oil droplet deposition was exhibited during NaCl/DI water cycles. Reduction of surface coverage occurred each time a highly concentrated sodium chloride solution was added to the system, regardless of the filtration flow regime (i.e., crossflow or dead-end). The observed decrease in the surface coverage was the desired evidence for PNIPAM chain conformational changes. 

## Figures and Tables

**Figure 1 membranes-09-00168-f001:**
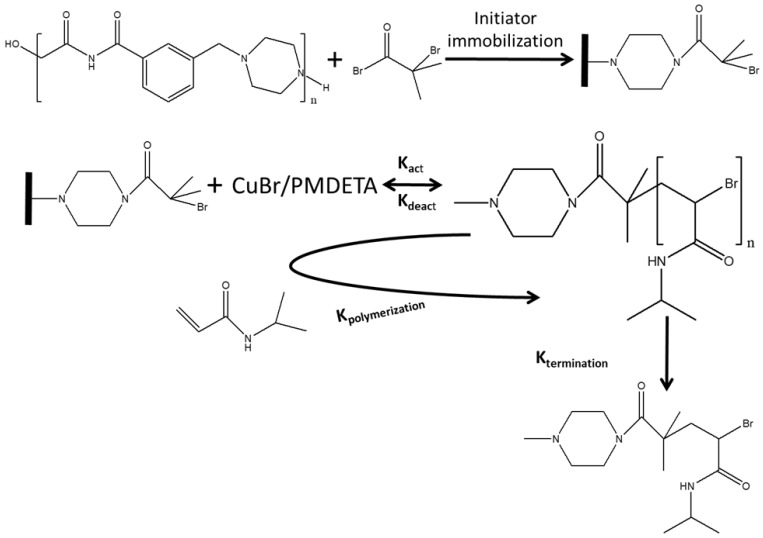
Schematic presentation of the modification process.

**Figure 2 membranes-09-00168-f002:**
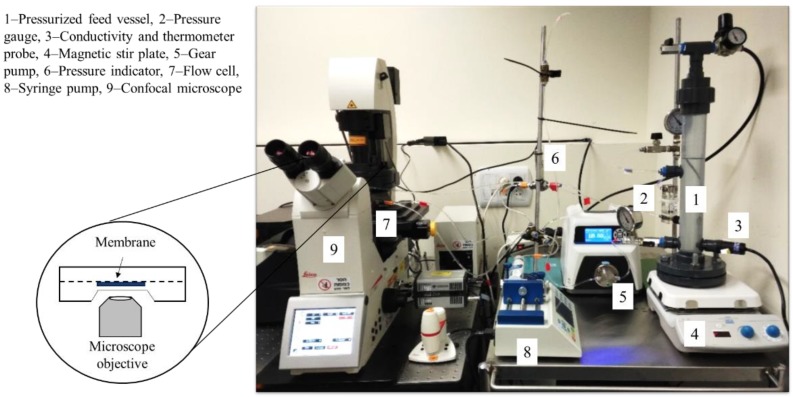
Direct microscopic observation experimental setup.

**Figure 3 membranes-09-00168-f003:**
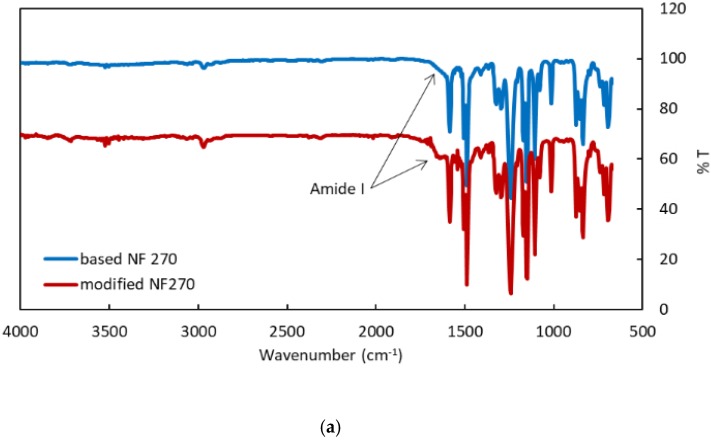
Membrane surface analysis of the base and modified membranes: (**a**) FTIR spectra; (**b**) water contact angle; (**c**) zeta potential.

**Figure 4 membranes-09-00168-f004:**
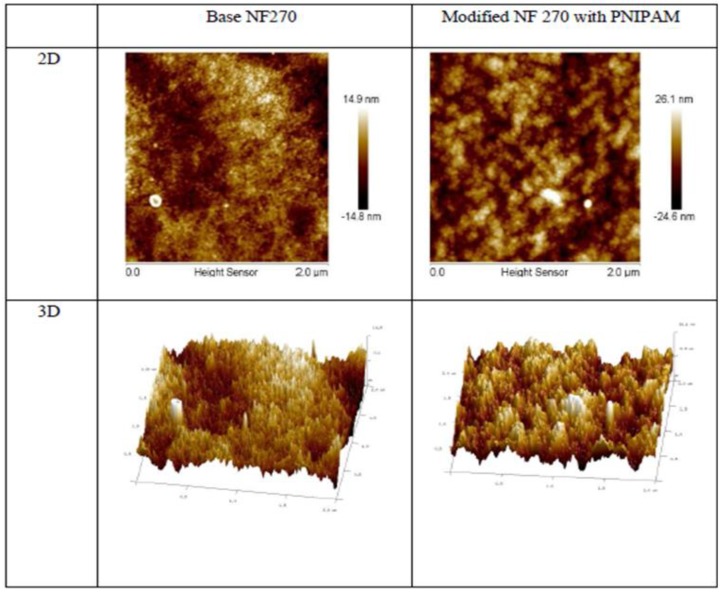
The 2D and 3D AFM images of the base and modified membranes. PNIPAM: poly(*N*-isopropylacrylamide).

**Figure 5 membranes-09-00168-f005:**
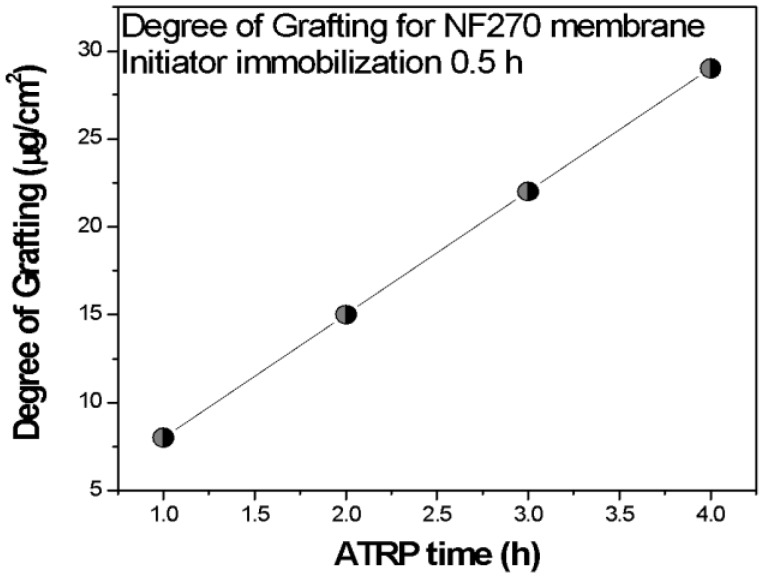
The degree of grafting of poly(NIPAM) as a function of atom transfer radical polymerization (ATRP) time.

**Figure 6 membranes-09-00168-f006:**
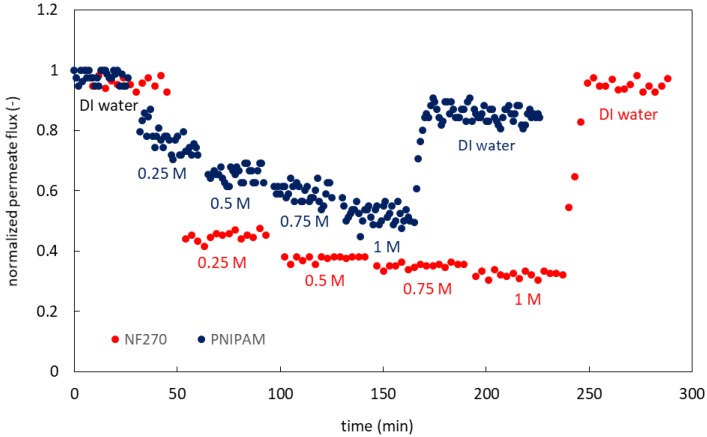
The variation of permeate flux as a function of time for different sodium chloride concentrations; the base NF 270 membrane is in red and the PNIPAM-modified membrane is in blue. Each time-step represents an increase of the NaCl concentration following this series: 0.25, 0.5, 0.75, and 1 M.

**Figure 7 membranes-09-00168-f007:**
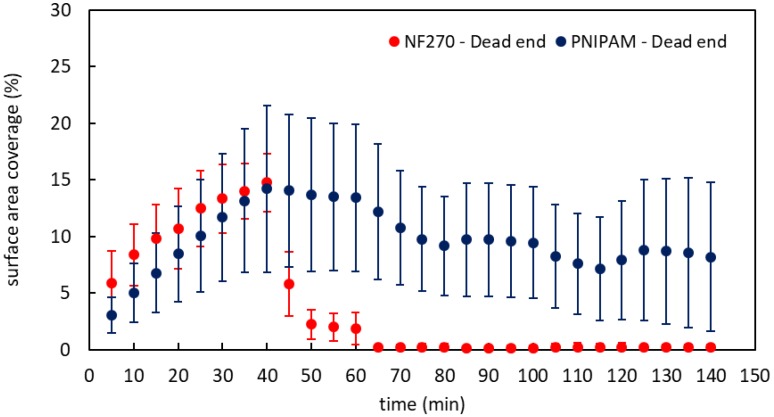
Surface coverage of oil droplets during dead-end filtration for base NF 270 (**red**) and PNIPAM-modified (**blue**) membranes. Oil droplets flux: NF 270 8.8 × 10^−13^ µl(oil)/s·µm^2^, PNIPAM 2.4 × 10^−13^ µl(oil)/s·µm^2^.

**Figure 8 membranes-09-00168-f008:**
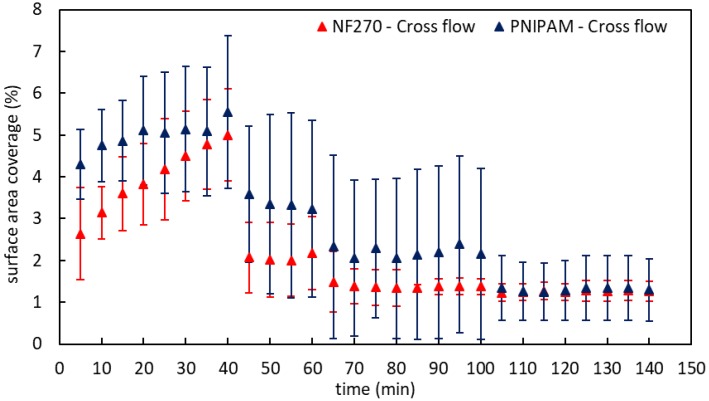
Surface area coverage by oil droplets during crossflow filtration for base NF 270 (**red**) and PNIPAM-modified (**blue**) membranes. Oil droplets flux 2.7 ×·10^−13^ µl(oil)/s·µm^2^.

**Figure 9 membranes-09-00168-f009:**
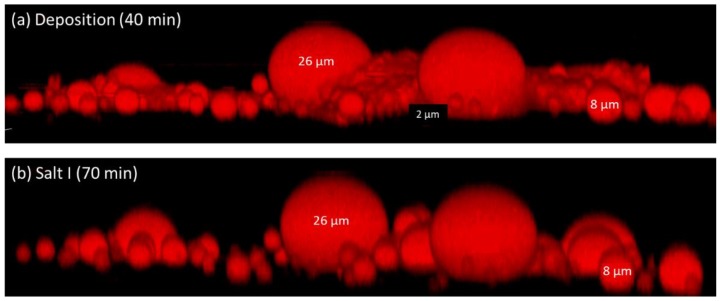
3D microscopic image of oil droplets deposited on PNIPAM-modified membrane at two different stages: (**a**) end of deposition and (**b**) first exposure to 1.5 M NaCl.

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
