# Peer review of "Oil Deposition on Polymer Brush-Coated NF Membranes"

_membranes, 2019, doi:10.3390/membranes9120168_

Round 1

Reviewer 1 Report

The membrane fouling due to deposition of oil droplets on the membrane surface compromises performance. The authors have real-time observation of the deposition of oil droplets by direct confocal microscopy. This work is a good idea and interesting research. The manuscript deserves to be published in the Journal after a Major Revision by considering the following comments:
1. The SEM images of base and modified membranes should be provided.
2. In Figure 2, the photo is not clear.
3. Why add gray background in Figure 3 and Figure 4.
4. The analytical results of the modified membrane and scientific explanation regarding their performances compared to the benchmark should be added and discussed.
5. State of the art references are missing in the paper. Such as Sep. Purif. Technol. 209 (2019) 434-442, J. Mater. Chem. A 7 (2019) 8491-8502, J. Colloid Interf. Sci 533 (2019) 278-286, J. Membrane. Sci. 573 (2019) 226-233, J. Membrane Sci. 595 (2020) 117499.

Reviewer 2 Report

The authors propose NF270 nanofiltration membranes modified by grafting poly(N-18 isopropylacrylamide) chains in order to avoid membrane fouling due to deposition of oil droplets on the membrane surface. In addition the indicate that switching conformation of the grafted polymer chains by using feed streams containing low and high NaCl concentrations can lead to partial release of adsorbed oil. The overall presentation contains some solid ideas but the manuscript has several flaws and is rather incomplete. Thus a major revision is recommended. More specifically:

Figure 1: Please bigger fonts for the elements are needed. Grey background does not improve clarity. I imagine that something went wrong with the conversion to pdf file. In addition the reaction scheme is confusing. Inclusion of triethylamine, 4-dimethylaminopyridine in the upper part of the arrow and acetonitrile in the lower part of the arrow would be very helpful. I assume the black rectangular stands for the part of the chemical formula of the membrane that is omitted. This must be indicated in the figure or in the legend.

Figure 2 is too dark and inacceptable. Please replace.

Please use a separated bigger figure for the IR spectra. The arrow is enough. Please remove the ellipse which partially covers the peak at 1635 cm-1. The latter can indeed be assigned to the bending of an amide group BUT there are two DIFFERENT amide functionalities most of all the N-H stretching vibration of the isopropyl amide group is missing (together with all the other characteristic peaks of PNIPAM). The fingerprint region (900-400 cm-1) of both spectra is cut off! Thus the authors do not provide adequate spectroscopy evidence for the claimed PNIPAM grafting. Inclusion of the 900-400 cm-1 area, two clear spectra in WHITE background a table of the peaks and a proper assignment are needed.      

I don’t understand what Cu(1)/Cu(2) stands for.

Since the authors performed experiments with polymerization time they must indicate this in the experimental section. (Polymerization was conducted for 1 to 4 hours). In addition I can’t understand why they did not test all modified membranes to find out the best and extract conclusions on the effect of the degree of grafting to the membrane performance. They need to do it though…    

Page 11 Lines 319 and 330: Most probably something went wrong with the reference management software.

Reviewer 3 Report

The manuscript describes the fouling of modified NF membranes with oil emulsions. The modification uses the well-known PNIPAM and its property changes caused by its lower critical solution temperature. While PNIPAM was already studied in a number of publications, the work described here seems to be novel and offers a new possible application. Direct observations of the fouling process are also of high interest, as the initial fouling is one the crucial factors during filtration.

The introduction is well written and gives a good overview about the research topic. Nevertheless, the reviewer noticed that roughly half of the cited literature has contribution of the corresponding author of this manuscript. It might be better to add a few works of other authors (e.g. Abetz et al.) to broaden the overview of this topic.

Section 3.2, first paragraph: This paragraph is confusing, as it describes colloidal fouling instead of the expected oil droplets. If the authors are referring to prior work, it should be cited at that point. If the actual work was described here, the text should be changed. (The difference between colloidal fouling and oil was emphasized in the manuscript a number of times.)

Regarding the dead-end filtration experiments: Why is the surface coverage decreasing for PNIPAM at the NaCl washing steps? An explanation should be given the description of fig.7, which might be supported be the microscopic images shown later in the manuscript. The same could be applied for the cross-flow experiments described afterwards.

Regarding the cleaning protocol for the dead-end experiments: It is stated that there is no TMP during the cleaning with NaCl solution/DI water. The reviewer assumes that the cleaning was thus carried out in cross-flow mode? If this is correct, it should be stated at least in the experimental section.

Another question might be how stable the switching process of PNIPAM is. Is there a limit of how often the polymer chains can be swollen/collapsed? This issue is of importance to possible applications and should be addressed either by suitable experiments or citation of prior work concerning this problem.

Minor issues/spell check:

Whole manuscript: PMDETA – abbreviation should be used instead of the full name (as it was done in fig.1).

Line 22 (abstract): Error: “chins”, needs to be changed into “chains”.

Line 77 Redundant period “Lin and Rutledge. [20] used “.

Line 80 XDLVO instead of „XDLCO“.

Line 126 “were” instead of “was” + missing period.

Figure 1: Final product is missing the brackets “[]n”.

Line 167 Sentence does not start with capital letter “oil”.

Line 238 Oxidation states should be given as Roman numbers: “Cu(I)/Cu(II)” instead of “Cu(1)/Cu(2)”.

Line 257 “salt concentration increases the LCST until it is below room temperate “ There might be a word missing here? “salt concentration increases, the LCST decreases until it is below room temperate”

Line 289 Redundant period after bracket “second exposure (105-120 min) .i.e.”.

Line 297 The last period is superscripted.

Line 311 A word is missing in this sentence. “the increase in osmotic back pressure“

Line 313 “are” instead of “is”.

Line 319 Seems like a citation is missing here. “Error! Reference source not found.”

Line 330 Same problem as in line 319.

Round 2

Reviewer 1 Report

all the mistakes have been corrected

Reviewer 2 Report

It is OK just use a thinner line for the IR spectra